# Dynamic Embeddings of Temporal High-Order Interactions via Neural Diffusion-Reaction Processes

## Abstract

High-order interactions of multiple entities are ubiquitous in practical applications. The associated data often includes the participants, interaction results, and the timestamps when each interaction occurred. While tensor factorization is a popular tool to analyze such data, it often ignores or underuses the valuable timestamp information. More important, standard tensor factorization only estimates a static representation for each entity, and ignores the temporal variation of the representations. However, such variations might reflect important evolution patterns of the underlying properties of the entities. To address these limitations, we propose Dynamical eMbedIngs of TempoRal hIgh-order interactions (DMITRI). We develop a neural diffusion-reaction process model to estimate the dynamic embeddings for the participant entities. Specifically, based on the observed interactions, we build a multi-partite graph to encode the correlation between the entities. We construct a graph diffusion process to co-evolve the embedding trajectories of the correlated entities, and use a neural network to construct a reaction process for each individual entity. In this way, our model is able to capture both the commonalities and personalities during the evolution of the embeddings for different entities. We then use a neural network to model the interaction result as a nonlinear function of the embedding trajectories. For model estimation, we combine ODE solvers to develop a stochastic mini-batch learning algorithm. We propose a simple stratified sampling method to balance the cost of processing each mini-batch so as to improve the overall efficiency. We show the advantage of our approach in both the ablation study and real-world applications.

## 1 Introduction

Many real-world applications are about interactions of multiple entities. For example, online shopping and promotion activities are interactions among *customers*, *commodities* and *online merchants*. A commonly used tool to analyze these high-order interactions is tensor factorization, which places the participant entities/objects in different tensor modes (or dimensions), and considers the interaction results as values of the observed tensor entries. Tensor factorization estimates an embedding representation for each entity, with which to reconstruct the observed entries. The learned embeddings can reflect the underlying structures within the entities, such as communities and outliers, and can be used as effective features for predictive tasks, such as recommendation and ads auction.

Practical data often includes the timestamps when each multiway interaction occurred. These timestamps imply rich, complex temporal variation patterns. Despite the popularity of tensor factorization, current methods often ignore the timestamps, or simply bin them into crude time steps (*e.g.*, by weeks or months) and jointly estimate embeddings for the time steps (Xiong et al., 2010; Rogers et al., 2013; Zhe et al., 2016a; 2015; Du et al., 2018). Therefore, the current methods might severely under-use the valuable temporal information in data. More important, standard tensor factorization always estimates a static embedding for each entity. However, as the representation of entities, these embeddings summarize the underlying properties of the entities, and can naturally evolve along with time, such as customer interests and preferences, user income and health, product popularity, and fashion. Learning static embeddings can miss capturing these interesting, important temporal knowledge.

To address these issues, we propose DMITRI, a dynamic embedding approach for temporal high-order interactions. We construct a nonlinear diffusion-reaction process in an Ordinary Differential Equation (ODE) framework to jointly estimate embedding trajectories for the participant entities. The ODE framework is known to be flexible and convenient to handle irregularly sampled timestamps and sparsely observed data (Rubanova et al., 2019), which is often the case in practice. In addition, since ODE models focus on learning the dynamics (*i.e.*, time derivatives) of the target function, they have promising potential for providing robust, accurate long-term predictions (via integration with the dynamics). Specifically, to leverage the structural knowledge within the data, we first build a multi-partite graph based on the observed interactions. The graph encodes the correlations between different types of entities in terms of their interaction history. We then construct a graph diffusion process in the ODE to co-evolve the embedding trajectories of correlated entities. Next, we use a neural network to construct a reaction process to model the individual-specific evolution for each entity. In this way, our neural diffusion-reaction process captures both the commonalities and personalities of the entities in learning their dynamic embeddings. Given the embedding trajectories, we model the interaction result as a latent function of the participants' trajectories. We use another neural network to flexibly estimate the function and to capture the complex relationships of the participant entities. For efficient training, we base on ODE solvers to develop a stochastic mini-batch learning algorithm. We develop a simple stratified sampling scheme, which can balance the cost of executing the ODE solvers in each mini-batch so as to improve the efficiency.

We evaluated our method in both simulation and real-world applications. The simulation experiments show that DMITRI can successfully capture the underlying dynamics of the entities from their temporal interactions, and recover the hidden clustering structures within the trajectories. Then in three real-world applications, we tested the accuracy in predicting the interaction results at different time points. DMITRI consistently outperforms the state-of-the-art tensor factorization methods that incorporate temporal information, often by a large margin. We also demonstrated that both the diffusion and reaction processes contribute to the learning and predictive performance. Finally, we investigated the learned embedding trajectories and found interesting evolution paths.

## 2 Notations and Background

Suppose we have collected data of interactions results between $K$ types of entities (*e.g.*, *customers*, *commodities* and *merchants*). Each type $k$ includes $d_k$ entities, and we index these entities by $1, \ldots, d_k$. We then index each interaction by a tuple $\boldsymbol{\ell} = (l_1, \ldots, l_K)$ where for each $k$, we have $1 \leq l_k \leq d_k$. Suppose we observed $N$ interactions, their results and timestamps. The dataset is denoted by $\mathcal{D} = \{(\boldsymbol{\ell}_1, t_1, y_1), \ldots, (\boldsymbol{\ell}_N, t_N, y_N)\}$ where $\{t_n\}$ and $\{y_n\}$ are the timestamps and interaction results. Our goal is for each entity $j$ of each type $k$, to estimate a dynamic embedding $\mathbf{u}_j^k(t) : \mathbb{R}_+ \to \mathbb{R}^R$. That is, the embedding is a time function (trajectory) of $R$-dimensional outputs.

High-order interaction data can be organized as multidimensional arrays or tensors. For example, we can create a $K$-mode tensor, and place the entities of type $k$ in mode $k$. Each interaction $\boldsymbol{\ell}$ is considered as an entry of the tensor, and the interaction result as the entry value. Hence, tensor factorization is a popular approach to process and analyze high-order interaction data. Standard tensor factorization introduces a static embedding representation for each entity, namely, $\mathbf{u}_j^k$ is considered as time invariant. Tensor factorization aims to estimate the embeddings (or factors) to reconstruct the tensor. For example, the classical Tucker decomposition (Tucker, 1966) employs a multilinear factorization model, $\mathcal{M} = \mathcal{W} \times_1 \mathbf{U}^1 \times_2 \ldots \times_K \mathbf{U}^K$, where $\mathcal{M} \in \mathbb{R}^{d_1 \times \ldots \times d_k}$ is the entire tensor, $\mathcal{W} \in \mathbb{R}^{R_1 \times \cdots \times R_K}$ is the tensor-core parameter, $\mathbf{U}^k$ comprises all the embeddings of the entities in mode $k$, and $\times_k$ is the tensor-matrix multiplication at mode $k$ (Kolda, 2006). The popular CANDECOMP/PARAFAC (CP) decomposition (Harshman, 1970) can be viewed as a simplified version of Tucker decomposition, where we set $R_1 = \ldots = R_K = R$ and the tensor-core $\mathcal{W}$ to be diagonal. Hence, each entry value is factorized as $m_{\boldsymbol{\ell}} = (\mathbf{u}_{l_1}^1 \circ \ldots \circ \mathbf{u}_{l_K}^K)^\top \boldsymbol{\lambda}$, where $\circ$ is the Hadamard (element-wise) product, and $\boldsymbol{\lambda}$ corresponds to $\mathrm{diag}(\mathcal{W})$. While CP and Tucker decomposition are popular and elegant, their multilinear modeling can be oversimplistic for complex applications. To estimate nonlinear relationships of the entities, Xu et al. (2012); Zhe et al. (2015; 2016a) used a Gaussian process (GP) (Rasmussen and Williams, 2006) to model the entry value as a random function of the embeddings, $m_{\boldsymbol{\ell}} = g(\mathbf{u}_{l_1}^1, \ldots, \mathbf{u}_{l_K}^K)$, where $g \sim \mathcal{GP}(0, \kappa(\mathbf{x}_{\boldsymbol{\ell}}, \mathbf{x}_{\boldsymbol{\ell}'}))$, $\mathbf{x}_{\boldsymbol{\ell}} = \{\mathbf{u}_{l_1}^1, \ldots, \mathbf{u}_{l_K}^K\}$ and $\mathbf{x}_{\boldsymbol{\ell}'} = \{\mathbf{u}_{l_1'}^1, \ldots, \mathbf{u}_{l_K'}^K\}$ are the embeddings of the entities in entry $\boldsymbol{\ell}$ and $\boldsymbol{\ell}'$, respectively, and $\kappa(\cdot, \cdot)$ is the covariance (kernel) function. Given the GP prior, any finite set of $N$ entry values follow a multi-variate Gaussian distribution, $\mathbf{m} \sim \mathcal{N}(\mathbf{0}, \mathbf{K})$, where $\mathbf{m} = [m_{\boldsymbol{\ell}_1}, \ldots, m_{\boldsymbol{\ell}_N}]$, $\mathbf{K}$ is the $N \times N$ kernel matrix, and each $[\mathbf{K}]_{i,j} = \kappa(\mathbf{x}_{\boldsymbol{\ell}}, \mathbf{x}_{\boldsymbol{\ell}'})$. Suppose we have collected continuous observations

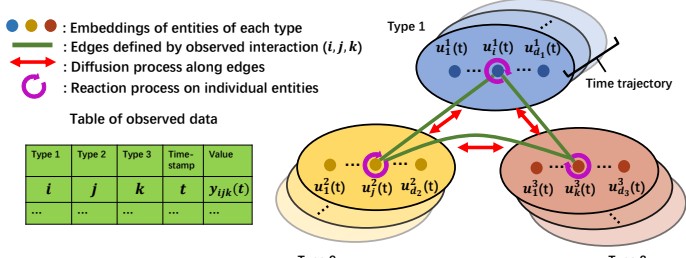

Figure 1: The illustration of the embedding model in DMITRI.

for the $N$ entries $\mathbf{y} = [y_1, \ldots, y_N]$. We can use a Gaussian noise model: $y_n = m_{\boldsymbol{\ell}_n} + \epsilon_n$ where $\epsilon_n \sim \mathcal{N}(0, \sigma^2)$. The marginal likelihood of the observations is $p(\mathbf{y}) = \mathcal{N}(\mathbf{y}|\mathbf{0}, \mathbf{K} + \sigma^2 \mathbf{I})$. We can maximize the likelihood to estimate the model parameters.

Practical interaction data often includes temporal information, *i.e.*, the timestamp when each observed interaction occurred. To integrate this information, current methods often bin the timestamps into a series of steps, say, by weeks or months (Xiong et al., 2010; Rogers et al., 2013; Zhe et al., 2016a; Song et al., 2017). The tensor is then expanded with an additional time-step mode. We can apply an arbitrary tensor factorization algorithm to estimate embeddings for both the entities and time steps. To learn the temporal dependency between the steps, a conditional model is often used (Xiong et al., 2010), say, $p(\mathbf{t}_{j+1}|\mathbf{t}_j) = \mathcal{N}(\mathbf{t}_{j+1}|\mathbf{t}_j, \tau\mathbf{I})$ where $\mathbf{t}_j$ is the embedding of $j$-th step. To leverage the continuous time information, Zhang et al. (2021) recently developed continuous CP decomposition, where the coefficients $\boldsymbol{\lambda}$ are modeled as a time function with polynomial splines.

## 3 Model

While successful, current tensor factorization methods assume the embeddings are static and time-invariant. However, the embeddings essentially summarize/extract the properties of entities to give a representation, and these properties can often evolve with time, such as customer interests, health status, and product popularity. Therefore, only estimating static embeddings can miss capturing important temporal variations of the entities' properties, resulting in poor representations and predictive performance. To address this issue, we propose DMITRI, a novel dynamic embedding approach.

Specifically, we propose an ODE model to learn the embedding trajectories $\{\mathbf{u}_j^k(t)|1 \le k \le K, 1 \le j \le d_k\}$. The ODE framework is known to be amenable for irregularly sampled, sparsely observed data, which is often the case in practice. More important, ODE models concentrate on learning the time derivative $\mathrm{d}\mathbf{u}_j^k/\mathrm{d}t$ (*i.e.*, dynamics), rather than the trajectory function itself. Therefore, they have a promising potential to give reliable, long-term trajectory prediction (via numerical integration) even at time points far away from the training timestamps, provided the time derivative is well captured. We construct a joint ODE model for all the embedding trajectories. The ODE consists of a diffusion process and a reaction process. The diffusion process leverages the structural knowledge in data to co-evolve the embeddings of correlated entities , so as to better overcome the data sparsity. The reaction process models the entity-specific evolution so that it can capture the individual differences in the embedding evolution. The ODE model synergizes the two processes to capture both the commonalities and personalities of these embedding trajectories.

**Diffusion Process on Multi-Partite Graphs.** First, we construct a graph-based diffusion process to exploit the entity correlations reflected in data $\mathcal{D}$. Intuitively, if a $K$-way interaction involves entity A (*e.g.*, customer A) and B (*e.g.*, commodity B), the two entities are likely correlated. Thus, we can draw an edge between A and B to express the correlation. We can then generalize this intuition to create a $K$-partite (undirected) graph $\mathbb{G}(E, V)$, to encode all such correlations across all the entities in the data. Each vertex represents a particular entity, and the entire collection of the entities is partitioned into $K$ groups, $V = V^1 \cup \ldots \cup V^K$, where group $V^k = \{v_1^k, \ldots, v_{d_k}^k\}$ represents the entities of type $k$. Two entities (of different types) are connected if they were observed to interact, namely, $(v_j^k, v_{j'}^{k'}) \in \mathbb{E}$ if there is some interaction $\boldsymbol{\ell}_n$ in $\mathcal{D}$ such that $\boldsymbol{\ell}_n = (\ldots, j, \ldots, j', \ldots)$ where $j$ and $j'$ are the indices of $k$-th and $k'$-th participants, respectively. That is, $l_{n_k} = j$ and $l_{n_{k'}} = j'$. See Fig. 1 for an example. This graph can naturally imply some underlying information diffusion across the entities within their interactions. For example, if customer A connects to products B and C, it might mean that A distributes their interests/willingness/budges to purchase B and C. The edges

between one merchant A and a list of products {B, C, ... } might indicate the diffusion of willingness to increase the inventory of these products.

To flexibly estimate the diffusion rate, we introduce a weight $w_{j,j'}^{k,k'}$ for each edge $(v_j^k, v_{j'}^{k'}) \in E$. We can then arrange these weights into $K(K+1)/2$ adjacent matrices, $\mathcal{W} = \{\mathbf{W}^{k,k'} | 1 \leq k, k' \leq K, k \neq k'\}$. Each $\mathbf{W}^{k,k'}$ is a sparse $d_k \times d_{k'}$ matrix that represents the edges and edge weights between $V^k$ and $V^{k'}$, i.e., $[\mathbf{W}^{k,k'}]_{j,j'} = w_{j,j'}^{k,k'}$ if $(v_j^k, v_{j'}^{k'}) \in E$ and 0 otherwise. We now construct a diffusion process based on the $K$-partite graph. We view the embedding trajectory as a kind of concentration. For each entity $j$ of type $k$, the change rate of its concentration (embedding) $\mathbf{u}_j^k(t)$ is determined by the difference from the concentrations of its neighbors. Since the neighbors can come from entities of all the other $K-1$ types, we have

$$\frac{\mathrm{d}\mathbf{u}_j^k}{\mathrm{d}t} = \sum_{s \in \{1,...,K\}\backslash k} \sum_{j'=1}^{d_s} [\mathbf{W}^{k,s}]_{j,j'} \left(\mathbf{u}_{j'}^s(t) - \mathbf{u}_j^k(t)\right) = \sum_{s \in \{1,...,K\}\backslash k} \left(\mathbf{w}_j^{k,s} \mathbf{U}^s(t)\right)^\top - a_j^{k,s} \mathbf{u}_j^k,$$

where $\mathbf{w}_j^{k,s}$ is the $j$-th row of $\mathbf{W}^{k,s}$, $\mathbf{U}^s(t) = [\mathbf{u}_1^s(t), \ldots, \mathbf{u}_{d_s}^s(t)]^\top$ is the embeddings of all the entities of type $s$, of size $d_s \times R$, and $a_j^{k,s} = \sum_{j'=1}^{d_s} [\mathbf{W}^{k,s}]_{j,j'}$ is the degree of vertex $j$ in $\mathbf{W}^{k,s}$. Consider $\mathbf{U}^k(t)$ — the embeddings of all the entities of type $k$, we therefore have

$$\frac{\mathrm{d}\mathbf{U}^k(t)}{\mathrm{d}t} = \sum_{s \in \{1...K\}\backslash k} \mathbf{W}^{k,s} \mathbf{U}^s(t) - \mathbf{A}^{k,s} \mathbf{U}^k(t) \tag{1}$$

where $\mathbf{A}^{k,s} = \mathrm{diag}(a_1^{k,s}, \ldots, a_{d_k}^{k,s})$ is the degree matrix of $\mathbf{W}^{k,s}$. We can see that the evolution of the embeddings for different types of entities are coupled. Hence, it is natural to formulate the diffusion process jointly for all the embeddings,

$$\frac{\mathrm{d}\mathcal{U}(t)}{\partial t} = \mathrm{d}\begin{pmatrix} \mathbf{U}^1(t) \\ \vdots \\ \mathbf{U}^K(t) \end{pmatrix} / \mathrm{d}t = \mathcal{W}\mathcal{U}(t) - \mathcal{A}\mathcal{U}(t) = (\mathcal{W} - \mathcal{A})\mathcal{U}(t) \tag{2}$$

where

$$\mathcal{W} = \begin{pmatrix} \mathbf{0} & \mathbf{W}^{1,2} & \cdots & \mathbf{W}^{1,K} \\ \mathbf{W}^{2,1} & \mathbf{0} & \cdots & \vdots \\ \vdots & & \ddots & \mathbf{W}^{K-1,K} \\ \mathbf{W}^{K,1} & \cdots & \mathbf{W}^{K,K-1} & \mathbf{0} \end{pmatrix}, \quad \mathcal{A} = \mathrm{diag}\begin{pmatrix} \sum_{s \in \{1...K\}\backslash 1} \mathbf{A}^{1,s} \\ \vdots \\ \sum_{s \in \{1...K\}\backslash k} \mathbf{A}^{k,s} \\ \vdots \\ \sum_{s \in \{1...K\}\backslash K} \mathbf{A}^{K,s} \end{pmatrix}.$$

**Reaction Process of Individual Entities.** Next, to capture the individual difference of each entity in evolving their embeddings, we model a local reaction process for each entity, $\mathbf{f}_{\boldsymbol{\theta}_k}(\mathbf{u}_j^k(t), t)$, where $\mathbf{f}(\cdot)$ is a neural network (NN), and $\boldsymbol{\theta}_k$ are the NN (reaction) parameters for type-$k$ entities[1]. The metaphor from the chemical physics is as follows. While substances are being diffused across different sites, at each site a chemical reaction process happens concurrently, which varies the concentration locally. We extend the model as

$$\frac{\mathrm{d}\mathbf{u}_j^k}{\mathrm{d}t} = \sum_{s \in \{1,...,K\}\backslash k} \sum_{j'=1}^{d_s} [\mathbf{W}^{k,s}]_{j,j'} \left(\mathbf{u}_{j'}^s(t) - \mathbf{u}_j^k(t)\right) + \mathbf{f}_{\boldsymbol{\theta}_k}(\mathbf{u}_j^k, t). \tag{3}$$

Our joint diffusion-reaction ODE model is therefore specified as follows,

$$\frac{\partial \mathcal{U}(t)}{\partial t} = (\mathcal{W} - \mathcal{A})\mathcal{U}(t) + \mathcal{F}(\mathcal{U}, t), \quad \mathcal{U}(0) = \mathcal{U}_0, \tag{4}$$

where $\mathcal{F}(\mathcal{U}, t) = [\mathbf{f}_{\boldsymbol{\theta}_1}(\mathbf{u}_1^1, t), \ldots, \mathbf{f}_{\boldsymbol{\theta}_1}(\mathbf{u}_{d_1}^1, t), \ldots, \mathbf{f}_{\boldsymbol{\theta}_K}(\mathbf{u}_1^K, t), \ldots, \mathbf{f}_{\boldsymbol{\theta}_K}(\mathbf{u}_{d_K}^K, t)]^\top$.

---

[1]Note that while the reaction model is the same and each type of entities share the same set of reaction parameters, those entities will have different reaction results due to the difference in the input to $\mathbf{f}$.

**Interaction Result Generation.** Given the embedding trajectories, to obtain the interaction result $m_\ell$ at arbitrary time $t$, we model $m_\ell(t)$ as a function of the relevant embeddings at time $t$,

$$m_\ell(t) = g\left(\mathbf{u}_{l_1}^1(t), \ldots, \mathbf{u}_{l_K}^K(t)\right). \tag{5}$$

While one can follow (Xu et al., 2012; Zhe et al., 2016b) to assign a GP prior over $g(\cdot)$, the GP model needs to compute a giant kernel matrix over all the observed interaction results (see Sec. 2). It is computationally too expensive or infeasible when the number of observations is large. Hence one has to seek for complex sparse approximations. To avoid this problem, we model $g$ with another neural network, which is not only as flexible as GP, but is more scalable and convenient for computation. Since now, the input to $g(\cdot)$ consists of the trajectory values, which vary with time, our NN model for $g$ can flexibly capture the complex temporal relationship of the entities. We finally sample the observed interaction results with a Gaussian noise model, $p(\mathbf{y}|\mathbf{m}) = \mathcal{N}(\mathbf{y}|\mathbf{m}, \sigma^2 \mathbf{I})$ where $\mathbf{y} = [y_1, \ldots, y_N]^\top$ and $\mathbf{m} = [m_{\ell_1}(t_1), \ldots, m_{\ell_N}(t_N)]^\top$. We focus on real-valued data in this paper. However, it is straightforward to extend our approach to other types of data.

## 4 Model Estimation

We now present the model estimation algorithm. Given data $\mathcal{D} = \{(\ell_1, t_1, y_1), \ldots, (\ell_N, t_N, y_N)\}$, the joint probability of our model is

$$p(\boldsymbol{\beta}, \{\boldsymbol{\theta}_k\}, \mathbf{y}) = p(\boldsymbol{\beta}) \cdot \prod_{k=1}^{K} p(\boldsymbol{\theta}_k) \cdot \prod_{n=1}^{N} \mathcal{N}\left(y_n | g\left(\mathbf{u}_{l_{n_1}}^1(t_n), \ldots, \mathbf{u}_{l_{n_K}}^K(t_n)\right), \sigma^2 \mathbf{I}\right), \tag{6}$$

where $\boldsymbol{\beta}$ is the NN parameters for $g$, each $\boldsymbol{\theta}_k$ is the NN reaction parameters for type-$k$ entities, their prior $p(\boldsymbol{\beta})$ and $p(\boldsymbol{\theta}_k)$ is element-wise standard Gaussian, and $\mathbf{y} = [y_1; \ldots; y_N]$. To obtain the trajectory values in the Gaussian likelihood of each $y_n$, we need to solve the ODE in (4) to time $t_n$,

$$\mathcal{U}(t_n) = \text{ODESolve}(\mathcal{U}_0, 0, t_n, \Theta) \tag{7}$$

where $\Theta = \{\mathcal{W}, \boldsymbol{\theta}_1, \ldots, \boldsymbol{\theta}_K\}$ consists of the ODE parameters. Our goal is to estimate the ODE parameters $\Theta$, the initial state $\mathcal{U}_0$, the NN parameters $\boldsymbol{\beta}$, and the noise variance $\sigma^2$.

**Stratified Mini-Batch Sampling.** We use stochastic mini-batch optimization to maximize the log joint probability so as to estimate all the required parameters,

$$\mathcal{L} = \log p(\boldsymbol{\beta}, \{\boldsymbol{\theta}_k\}, \mathbf{y}) = \log(\text{Prior}) - \sum_{n=1}^{N} \log \mathcal{N}\left(y_n | g\left(\mathbf{x}_n\right), \sigma^2 \mathbf{I}\right)$$

where $\log(\text{Prior}) = \log p(\boldsymbol{\beta}) + \sum_{k=1}^{K} \log p(\boldsymbol{\theta}_k)$, and $\mathbf{x}_n = [\mathbf{u}_{l_{n_1}}^1(t_n), \ldots, \mathbf{u}_{l_{n_K}}^K(t_n)]$. Each time, we sample a mini-batch of interaction results $\mathcal{B}$, and obtain an unbiased stochastic estimate of the log probability, $\widehat{\mathcal{L}} = \log(\text{Prior}) + \frac{N}{B} \sum_{n \in \mathcal{B}} \left[\log \mathcal{N}(y_n | g(\mathbf{x}_n), \sigma^2)\right]$. We compute $\nabla \widehat{\mathcal{L}}$ as the stochastic gradient to update all the parameters.

For each data point $n$ in the mini-batch, we need to run ODE solving (7) to obtain $\mathbf{x}_n = [\mathbf{u}_{l_{n_1}}^1(t_n), \ldots, \mathbf{u}_{l_{n_K}}^K(t_n)]$. To back-propagate the gradient so as to compute the gradient w.r.t the ODE parameters $\Theta$ and initial state $\mathcal{U}_0$, we can either construct a computational graph during the running of the solver (*e.g.*, the Runge-Kutta method (Dormand and Prince, 1980)), or use the adjoint state method (Pontryagin, 1987; Chen et al., 2018) that solves an adjoint backward ODE to compute the gradient. In whichever case, we need to sort the time points in the mini-batch and solve the ODE sequentially for these time points. As a result, the number of *unique* time points in the mini-batch greatly influences the speed of processing the mini-batch. The standard mini-batch sampling (based on the training example indices) can result in an uneven allocation of the computational cost across the mini-batches — some mini-batch is fast and some including more time points is much slower. To address this issue, we use a simple stratified sampling approach.

- We collect the unique time points in the whole dataset, $\mathcal{T} = \{\tau_1, \tau_2, \ldots\}$ at the beginning.
- To conduct each stochastic update, we first sample $B$ unique time points $\mathcal{C}$ from $\mathcal{T}$, then for each time point $\tau_j \in \mathcal{C}$, we look at all the interactions occurred at $\tau_j$, namely $\mathcal{D}_{\tau_j} = \{(\ell_n, t_n, y_n) \in \mathcal{D} | t_n = \tau_j)\}$.
- We randomly sample one interaction example from each $\mathcal{D}_{\tau_j}$ to collect the mini-batch $\mathcal{B}$.

In this way, we ensure the cost of running ODE solvers and related gradient computation in each mini-batch is identical. There are no fluctuations in cost/running time when processing different min-batches. Empirically, we found that the overall speed of our method is much faster than vanilla stochastic mini-batch optimization.

## 5 Related Work

To integrate the temporal information, most tensor factorization approaches augment the tensor with a time mode (Xiong et al., 2010; Rogers et al., 2013; Zhe et al., 2016b; Ahn et al., 2021; Zhe et al., 2015; Du et al., 2018), which includes a list of time steps. They jointly estimate the embeddings for entities and time steps. To estimate the temporal dependencies, existing methods often employ a dynamic model over the time steps, such as a conditional Gaussian prior (Xiong et al., 2010), recurrent neural networks(Wu et al., 2019), and kernel smoothing/regularization (Ahn et al., 2021). To leverage continuous timestamps, Zhang et al. (2021) modeled the CP coefficients $\boldsymbol{\lambda}$ as a time function, and estimated it with polynomial splines. The most recent work (Wang and Zhe, 2022) places a GP prior in the frequency domain, and construct a bi-level GP model to learn factor trajectories as a combination of Fourier bases. Another set of works factorize the interaction events (Schein et al., 2015; 2016; Zhe and Du, 2018; Pan et al., 2020; Wang et al., 2020), rather than the interaction results (*e.g.*, purchase amount and product ratings). These works mainly leverage Poisson processes, Hawkes processes, or more general point processes to estimate the event rate. Like the standard tensor factorization, these methods also estimate static embeddings for the event participants.

Our model can be viewed as an extension of the neural ODE model (Chen et al., 2018). If we only employ the reaction process for each entity, our model is a latent neural ODE (we have an additional NN that combines the latent trajectories to predict the interaction results). However, we further leverage the structural knowledge in data to construct a multi-partite graph so as to encode the correlations of the participants in their high-order interactions. Based on the multi-partite graph, we construct a diffusion process to co-evolve the embeddings of the entities. In doing so, we can better overcome the data sparsity issue. Our work is also related to (Rubanova et al., 2019), which uses neural ODEs to model the state transition of recurrent neural networks (RNNs) or to build auto-regressive models for time series analysis. The major difference is that our work deals with time series of high-order interactions, and our ODE is used to model the evolution of the embeddings of the interaction participants (rather than the RNN states). Many other works have leveraged/developed graph diffusion processes given the graph data. For example, Chamberlain et al. (2021) proposed a graph neural network (GNN) by using multi-head attention to construct the adjacent matrix for a graph diffusion equation over the graph nodes. Atwood and Towsley (2016) introduced a diffusion operator to develop diffusion-convolutional neural networks. Huang et al. (2021) developed a GNN based ODE to model both the nodes and edges in dynamic graphs.

## 6 Experiment

### 6.1 Ablation Study

We first examined DMITRI on a synthetic task. Specifically, we considered interactions among two types of entities, where each type includes 20 entities. Each entity has one underlying embedding trajectory. For type-1, the trajectory of each entity is an exponential function, $u_j^1(t) = c_j^1 \exp(0.5 c_j^1 t)$ ($1 \leq j \leq 20$), while for type-2 is a linear function, $u_j^2(t) = c_j^2 + 2\pi c_j^2 t$. We generated two clusters of trajectories for each entity type, where those of the first ten entities form the first cluster and the remaining the second cluster. To this end, for type 1, we sampled the coefficients of the first ten entities' trajectories, namely, $[c_1^1, \ldots, c_{10}^1]^\top$, from $\mathcal{N}\left([-5, \ldots, -5]^\top, 0.1\mathbf{I}\right)$, and the remaining ten coefficients $[c_{11}^1, \ldots, c_{20}^1]^\top$ from $\mathcal{N}\left([0.5, \ldots, 0.5]^\top, 0.1\mathbf{I}\right)$. Then for type 2, we sampled each coefficient $c_j^2$ conditioned on its counter-part for type 1, $c_j^2|c_j^1 \sim \mathcal{N}(c_j^2|c_j^1, 0.1)$. That means, the coefficients of cluster-1 trajectories across the two event types are close, and so are those of cluster-2 trajectories. The result for a particular interaction $\boldsymbol{\ell} = (l_1, l_2)$ is generated by

$$m_{\boldsymbol{\ell}}(t) = \left(u_{l_1}^1(t)\right)^{\mathbb{1}(l_1+l_2 \mod 2=0)} \cdot \left(u_{l_2}^2(t)\right)^{\mathbb{1}(l_1+l_2 \mod 2=1)}. \tag{8}$$

where $\mathbb{1}(\cdot)$ is the indicator function. When $l_1 + l_2$ is even, the interaction result is the trajectory value of the first entity; otherwise, it is the trajectory value of the second entity. To generate the training data, we randomly sampled interactions from $\{(l_1, l_2)|1 \leq l_1, l_2 \leq 10\} \cup \{(l_1, l_2)|11 \leq l_1, l_2 \leq 20\}$ (namely, interactions between cluster-1 entities of the two types, and between cluster-2 entities). We then sampled $t \sim \text{Unifrom}[0, 5]$, to obtain the interaction results. We randomly generated 6,400 interaction results and the timestamps for training, and another 1,600 data points for testing.

We implemented our method with Pytorch (Paszke et al., 2019). We used `torchdiffeq` library (https://github.com/rtqichen/torchdiffeq) to solve ODEs and to compute the gradient w.r.t ODE parameters and initial states via automatic differentiation. For the NN of the

reaction process, we used one hidden layer, with 10 neurons and tanh activation, and for the NN to predict the interaction result, we used two hidden layers, 50 neurons per layer and tanh activation.

We compared with NONFAT (NONparametric Factor Trajectory learning) (Wang and Zhe, 2022), a bi-level latent GP model that uses Fourier bases to estimate factor trajectories for dynamic tensor decomposition. To our knowledge, this work is the only method (and also the most recent) that also estimates trajectories. We used the original implementation (`https://github.com/wzhut/NONFAT`) and the default settings. We set the mini-batch size to 50, and used ADAM (Kingma and Ba, 2014) algorithm for stochastic optimization. The learning rate was automatically adjusted in $[10^{-4}, 10^{-1}]$ by the `ReduceLROnPlateau` scheduler (Al-Kababji et al., 2022). The maximum number of epochs is 2K, which is enough for convergence. The estimated trajectories are shown in Fig. 2c-f. As we can see, our estimation (Fig. 2e and 2f) well matches the ground-truth and accurately recovers the cluster structure of the trajectories. The root-mean-square error (RMSE) on the test set is 0.032. By contrast, although the test error of NONFAT is close to DMITRI (0.034), its learned trajectories (Fig. 2c and 2d) are far from the ground-truth, and fail to reflect the cluster structure. This might be due to that GP models are difficult to capture the interaction function (8), which is a switch function. These have shown the advantage of DMITRI in capturing complex relationships within data to recover the underlying trajectories and their structure.

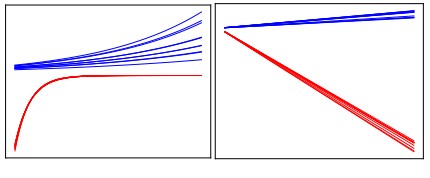

(a) Ground-truth: type 1 (b) Ground-truth: type 2

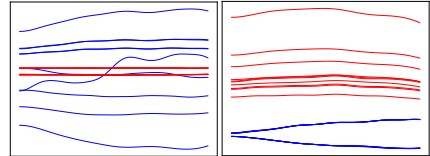

(c) NONFAT: Type 1 (d) NONFAT: Type 2

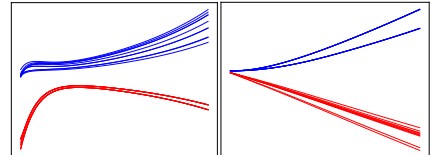

(e) DMITRI: Type 1 (f) DMITRI: Type 2

Figure 2: The estimated embedding trajectories for each entity type. The color indicates the ground-truth cluster membership

## 6.2 Prediction Accuracy

**Datasets.** We next evaluated the predictive performance of DMITRI in three real-world applications.

(1) *CA Weather* (Moosavi et al., 2019) (`https://smoosavi.org/datasets/lstw`), weather conditions in California from August 2016 to December 2020. We extracted four-way interactions among 7 different weather *types*, 6 *severity levels*, 30 *latitudes* and 30 *longitudes* in GPS coordinate. The interaction re-

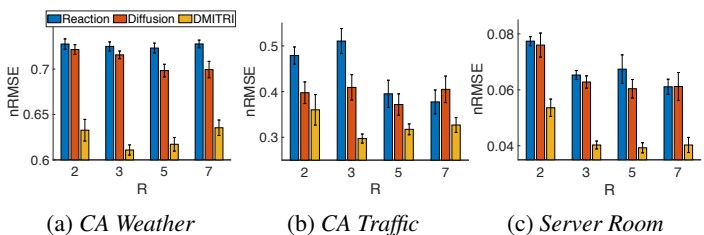

(a) *CA Weather* (b) *CA Traffic* (c) *Server Room*

Figure 3: Predictive performance of the diffusion and reaction processes

sult is the count of the particular weather condition. We collected 15K interactions and the timestamps. (2) *CA Traffic* (Moosavi et al., 2019) (`https://smoosavi.org/datasets/lstw`), traffic accidents in California from January 2018 to December 2020. We extracted four-way interactions between *traffic type*, *severity level*, *latitudes*, *longitude*. There are 7 traffic types, 6 severity levels, 20 latitudes and 20 longitudes. We collected 30K interaction results (accident counts) at different time points. (3) *Server Room* (`https://zenodo.org/record/3610078#.XlNpAigzaM8`), temporal temperature records of Poznan Supercomputing and Networking Center. The temperatures were measured at 34 locations, under different air-condition modes ($24°$, $27°$, and $30°$) and power usage settings (50%, 75% and 100%). Hence, we extracted three-way interactions (*location*, *air-condition mode*, *power level*). In total, 10K interactions and their timestamps were collected.

**Competing Methods.** The following popular and/or state-of-the-art temporal factorization approaches were compared. (1) CP-DTLD, discrete-time CP decomposition with linear dynamics, where a conditional prior is placed over successive time steps, $p(\mathbf{t}_{j+1}|\mathbf{t}_j) = \mathcal{N}(\mathbf{t}_{j+1}|\mathbf{A}\mathbf{t}_j + \mathbf{b}, v\mathbf{I})$; $\mathbf{A}, \mathbf{b}$ and $\mathbf{v}$ were jointly estimated during the CP decomposition. Note that (Xiong et al., 2010) is an instance of this model where $\mathbf{A} = \mathbf{I}$ and $\mathbf{b} = \mathbf{0}$. (2) GP-DTLD and (3) NN-DTLD, similar to CP-DTLD, except using GP (Zhe et al., 2016b) and NN decomposition models (similar to (Liu et al., 2019)), respectively.

| CA Weather | $R = 2$ | $R = 3$ | $R = 5$ | $R = 7$ |
|---|---|---|---|---|
| CP-DTLD | $0.7440 \pm 0.0035$ | $0.7372 \pm 0.0040$ | $0.7290 \pm 0.0042$ | $0.7270 \pm 0.0044$ |
| GP-DTLD | $0.7417 \pm 0.0031$ | $0.7414 \pm 0.0036$ | $0.7444 \pm 0.0036$ | $0.7449 \pm 0.0039$ |
| NN-DTLD | $0.7228 \pm 0.0054$ | $0.7116 \pm 0.0033$ | $0.7070 \pm 0.0041$ | $0.7065 \pm 0.0038$ |
| CP-DTND | $0.7448 \pm 0.0031$ | $0.7360 \pm 0.0035$ | $0.7273 \pm 0.0037$ | $0.7280 \pm 0.0044$ |
| GP-DTND | $0.7399 \pm 0.0034$ | $0.7346 \pm 0.0032$ | $0.7448 \pm 0.0037$ | $0.7467 \pm 0.0031$ |
| NN-DTND | $0.7113 \pm 0.0045$ | $0.6979 \pm 0.0126$ | $0.6659 \pm 0.0122$ | $0.6543 \pm 0.0155$ |
| CP-CT | $1.0000 \pm 0.0096$ | $0.9959 \pm 0.0067$ | $1.0010 \pm 0.0017$ | $1.0060 \pm 0.0034$ |
| GP-CT | $0.7433 \pm 0.0038$ | $0.7354 \pm 0.0027$ | $0.7359 \pm 0.0034$ | $0.7377 \pm 0.0033$ |
| NN-CT | $0.8697 \pm 0.0014$ | $0.8679 \pm 0.0022$ | $0.8676 \pm 0.0018$ | $0.8695 \pm 0.0016$ |
| NONFAT | $0.7444 \pm 0.0042$ | $0.7460 \pm 0.0032$ | $0.7645 \pm 0.0061$ | $0.7553 \pm 0.0029$ |
| DMITRI | $\mathbf{0.6327 \pm 0.0119}$ | $\mathbf{0.6109 \pm 0.0056}$ | $\mathbf{0.6172 \pm 0.0075}$ | $\mathbf{0.6354 \pm 0.0085}$ |
| CA Traffic | | | | |
| CP-DTLD | $0.6498 \pm 0.0257$ | $0.6424 \pm 0.0266$ | $0.6436 \pm 0.0268$ | $0.6405 \pm 0.0262$ |
| GP-DTLD | $0.6309 \pm 0.0167$ | $0.6290 \pm 0.0185$ | $0.6383 \pm 0.0204$ | $0.6496 \pm 0.0193$ |
| NN-DTLD | $0.6528 \pm 0.0230$ | $0.6545 \pm 0.0244$ | $0.6401 \pm 0.0282$ | $0.6136 \pm 0.0338$ |
| CP-DTND | $0.6497 \pm 0.0245$ | $0.6456 \pm 0.0265$ | $0.6431 \pm 0.0263$ | $0.6419 \pm 0.0259$ |
| GP-DTND | $0.6544 \pm 0.0213$ | $0.6559 \pm 0.0224$ | $0.6604 \pm 0.0243$ | $0.6674 \pm 0.0214$ |
| NN-DTND | $0.6578 \pm 0.0248$ | $0.6528 \pm 0.0256$ | $0.6519 \pm 0.0249$ | $0.6482 \pm 0.0261$ |
| CP-CT | $0.9858 \pm 0.0120$ | $0.9972 \pm 0.0056$ | $0.9816 \pm 0.0136$ | $0.9991 \pm 0.0120$ |
| GP-CT | $0.6610 \pm 0.0207$ | $0.6668 \pm 0.0191$ | $0.6756 \pm 0.0190$ | $0.6768 \pm 0.0196$ |
| NN-CT | $0.9804 \pm 0.0017$ | $0.9815 \pm 0.0015$ | $0.9791 \pm 0.0012$ | $0.9802 \pm 0.0017$ |
| NONFAT | $0.4461 \pm 0.0247$ | $0.4610 \pm 0.0231$ | $0.5031 \pm 0.0155$ | $0.6307 \pm 0.0847$ |
| DMITRI | $\mathbf{0.3601 \pm 0.0334}$ | $\mathbf{0.2972 \pm 0.0099}$ | $\mathbf{0.3174 \pm 0.0118}$ | $\mathbf{0.3269 \pm 0.0162}$ |
| Server Room | | | | |
| CP-DTLD | $0.4211 \pm 0.0029$ | $0.4209 \pm 0.0031$ | $0.4208 \pm 0.0028$ | $0.4208 \pm 0.0028$ |
| GP-DTLD | $0.0914 \pm 0.0020$ | $0.0791 \pm 0.0010$ | $0.0739 \pm 0.0014$ | $0.0753 \pm 0.0013$ |
| NN-DTLD | $0.4213 \pm 0.0032$ | $0.4213 \pm 0.0032$ | $0.4212 \pm 0.0034$ | $0.4205 \pm 0.0030$ |
| CP-DTND | $0.2835 \pm 0.0160$ | $0.1751 \pm 0.0020$ | $0.1174 \pm 0.0011$ | $0.0829 \pm 0.0044$ |
| GP-DTND | $0.0925 \pm 0.0013$ | $0.0784 \pm 0.0011$ | $0.0739 \pm 0.0009$ | $0.0774 \pm 0.0009$ |
| NN-DTND | $0.4213 \pm 0.0032$ | $0.4212 \pm 0.0030$ | $0.4211 \pm 0.0032$ | $0.4205 \pm 0.0030$ |
| CP-CT | $0.9919 \pm 0.0096$ | $0.9951 \pm 0.0050$ | $0.9862 \pm 0.0109$ | $1.0121 \pm 0.0070$ |
| GP-CT | $0.1385 \pm 0.0020$ | $0.1223 \pm 0.0016$ | $0.1275 \pm 0.0014$ | $0.1365 \pm 0.0014$ |
| NN-CT | $0.1193 \pm 0.0030$ | $0.1140 \pm 0.0015$ | $0.1113 \pm 0.0027$ | $0.1149 \pm 0.0028$ |
| NONFAT | $0.1468 \pm 0.0026$ | $0.1407 \pm 0.0023$ | $0.1396 \pm 0.0022$ | $0.1409 \pm 0.0030$ |
| DMITRI | $\mathbf{0.0536 \pm 0.0031}$ | $\mathbf{0.0403 \pm 0.0014}$ | $\mathbf{0.0393 \pm 0.0018}$ | $\mathbf{0.0403 \pm 0.0027}$ |

Table 1: Normalized Root Mean-Square Error (nRMSE). The results were averaged from five runs.

(4) CP-DTND, (5) GP-DTND and (6) NN-DTND — CP, GP and NN decomposition with nonlinear dynamics, where the conditional prior is $p(\mathbf{t}_{j+1}|\mathbf{t}_j) = \mathcal{N}(\mathbf{t}_{j+1}|\sigma(\mathbf{A}\mathbf{t}_j) + \mathbf{b}, v\mathbf{I})$ where $\sigma(\cdot)$ is an nonlinear activation. The dynamics can therefore be viewed as an RNN transition. (7) CP-CT (Zhang et al., 2021), continuous-time CP factorization, which models the CP coefficients as a time-varying function, with polynomial splines. (8) GP-CT, continuous-time GP decomposition that extends (Xu et al., 2012; Zhe et al., 2016b) by plugging the time in the GP kernel so as to estimate the interaction result as a function of the embeddings and time, $m_\ell = g(\mathbf{u}_{\ell_1}^1, \ldots, \mathbf{u}_{\ell_K}^K, t)$. (9) NN-CT, continuous-time NN decomposition, where the input consists of both the embeddings and time $t$. (10) NON-FAT (Wang and Zhe, 2022).

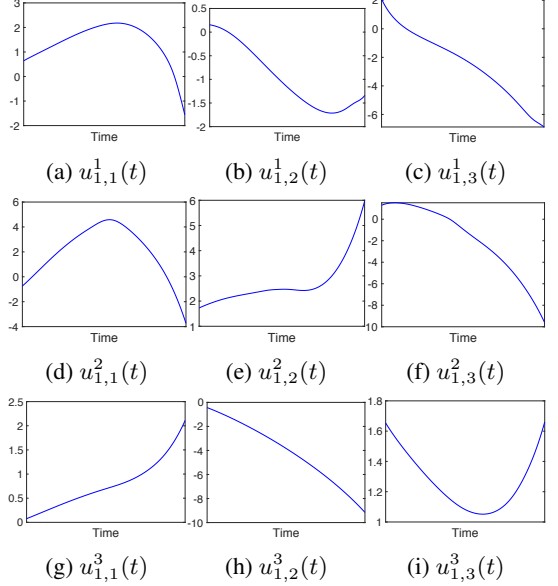

(a) $u_{1,1}^1(t)$    (b) $u_{1,2}^1(t)$    (c) $u_{1,3}^1(t)$

(d) $u_{1,1}^2(t)$    (e) $u_{1,2}^2(t)$    (f) $u_{1,3}^2(t)$

(g) $u_{1,1}^3(t)$    (h) $u_{1,2}^3(t)$    (i) $u_{1,3}^3(t)$

Figure 4: The learned embedding trajectories for location 1 (a-c), air conditional mode 1 (d-f), and power usage level 1 (g-i) in *Server Room* dataset.

**Settings and Results.** All the approaches were implemented with PyTorch. The Square Exponential kernel was used for all the GP related methods, including GP-{DTLD, DTND, CT}. We used the same variational sparse approximation (Hensman et al., 2013) to fulfill scalable posterior inference. Following (Zhe et al., 2016b), the number of inducing point was set to 100. For the NN decomposition methods, we employed a three-layer network with `tanh` activation. The layer width was chosen from {10, 25, 50, 75, 100}. We used `tanh` as the activation function in the nonlinear dynamic baselines, including {CP, GP, NN}-DTND. For our method, we used the same NN architecture for both the reaction process and interaction result prediction, which includes two hidden layers with 50 neurons per layer. For CP-CT, we employed 100 knots to fulfill the polynomial splines. For each discrete-time method, the number of time steps was chosen from {25, 50, 75, 100} via the cross-validation on the training set. We trained all the models with stochastic mini-batch optimization. We used the ADAM algorithm, and the mini-batch size was set to 100. We ran every method with 10K epochs to ensure convergence. The learning rate was automatically adjusted in $[10^{-4}, 10^{-1}]$ by the `ReduceLROnPlateau` scheduler. We varied the dimension of the embeddings $R$ from {2, 3, 5, 7}. For DMITRI, $R$ is the number of embedding trajectories; we used computational graphs to obtain the gradient. We followed (Xu et al., 2012; Kang et al., 2012; Zhe et al., 2016b) to randomly draw 80% observed interactions and their time stamps for training, with the remaining for test. We computed the normalized root-mean-square error (nRMSE). We repeated the evaluation for five times and computed the average nRMSE and standard deviation.

As shown in Table 1, DMITRI consistently achieves the best prediction accuracy, and in many cases outperforms the competing methods by a large margin. Although learning an embedding trajectory is much more challenging than learning a fixed-value embedding, the experimental results have demonstrated the advantage of our method in predictive performance. To investigate the effect of the two processes in our model, we also examined our method with the diffusion process only and reaction process only on all the datasets. Their predictive performance, as compared with DMITRI, is shown in Fig. 3. We can see that each individual component can lead to good prediction accuracy. However, each component is worse than DMITRI that synergizes the two components together. Therefore, the results show that each process is effective, and more important, the two processes can bolster each other to further improve the performance when they are combined.

### 6.3 Learning Result Investigation

Next, we looked into the learned embedding trajectories and checked if they exhibit patterns. To do so, we set $R = 3$ and ran DMITRI on *Server Room* dataset. In Fig. 4, we show the learned embedding trajectories for the first location (a-c), the first air condition mode (d-f) and the first power usage level (g-i). As we can see, even for the same object, *e.g.*, a particular location, the corresponding embedding trajectories vary quite differently, implying the evolution of different underlying properties, such as the workload, memory usage, and network latency.

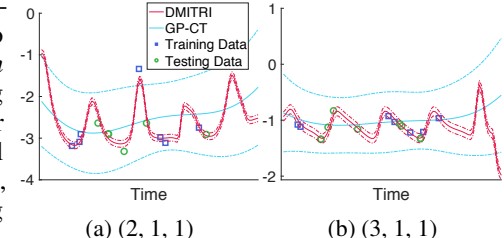

(a) (2, 1, 1)  (b) (3, 1, 1)

Figure 5: Interaction result prediction on *Server Room*.

Finally, we showcase the temporal predictions for two interactions. As we can see from Fig. 5, given only a few training points (blue), our method can predict the test points (green) much more accurately, as compared with GPCT, and the predictive uncertainty (reflected by the noise variance $\sigma^2$) is much smaller. This might be due to that via diffusion-reaction process, our method can more effectively extract the temporal knowledge from sparse data. For example, DMITRI successfully captured the periodic nature in the first interaction (Fig. 5a) while GPCT treated the fluctuation as noises and ended up with much inaccurate predictions and much larger predictive variances.

## 7 Conclusion

We have presented DMITRI, a neural diffusion-reaction process model to learn dynamic embeddings from high-order interaction time series. The predictive performance is encouraging and the learned embedding trajectories exhibit interesting patterns. Currently, our method is limited to a small number of entities since it has to integrate the entire multi-partite graph to construct the diffusion process. In the future work, we plan to develop graph cut algorithms to partition the graph into a set of small sub-graphs, so that we can construct multiple diffusion processes in parallel so as to scale up to big graphs and a large number of entities.

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
