# OpenReview forum: "Dynamic Embeddings of Temporal High-Order Interactions via Neural Diffusion-Reaction Processes"
_ICLR.cc/2023/Conference — Submitted to ICLR 2023_

### Official Review · Reviewer_eUMJ · 2022-10-14

**Confidence:** 2
**Correctness:** 3
**Technical Novelty And Significance:** 3
**Empirical Novelty And Significance:** 2
**Recommendation:** 5

**Clarity, Quality, Novelty And Reproducibility:**

Clarity:
The technical parts in Sec.3 about the diffusion reaction processes, are not that easy to digest, to me.
THe final experimental protocol is not entirely clear.

Quality:
Overall, the method looks solid.

Novelty:
The studied method seems to be novel in its motivated context.

**Strength And Weaknesses:**

Strength :

The paper targets an interesting question, and the method looks solid.

Weaknesses :

To me, the technical part of Sec. 3 is not that easy to digest. For instance, it is necessary to thoroughly explain why to construct the diffusion and reaction processes in this way, for someone else to take it up.

For the results in Table 1, it is entirely unclear that what does the RMSE error reflect? On pp.9, you only mention use remaining for test. But, I cannot find what are you predicting in the test set!

**Summary Of The Paper:**

This paper proposes a neural diffusion-reaction process model, which can take into account the timestamps and model the temporal embedding of participants behind high-order interactions, compared with existing related tensor factorization methods.

- To that end, they build the model upon the ODE framework that can handle continuously observed timestamps.
- They build a multi-partite graph on observed interactions, and learn a graph diffusion process to allow the dynamic embeddings of correlated participants to co-evolve over time.
- A separate neural net is used to construct a reaction process to capture individual difference.
- The final interaction result is captured using a Gaussian observation model, of which the mean is a function of the dynamic embedding.

**Summary Of The Review:**

This paper develops a new model to capture timestamps and to model dynamic embedding of entities behind high-order interaction time series, which cannot be well captured using existing tensor factorization solutions.

To me, the method looks interesting and solid, but somewhat novel in the motivated context.

In addition, the final experiment is not entirely clear to me.

---

> ### Author Response · Authors · 2022-11-13
> **Thanks for your comments and here is our clarification**
>
> Thanks for your valuable comments. Here is our clarification. C: comments; R: response.
>
> C1: To me, the technical part of Sec. 3 is not that easy to digest. For instance, it is necessary to thoroughly explain why to construct the diffusion and reaction processes in this way, for someone else to take it up.
>
> R1: We appreciate your feedback. We do agree that our technique part might be "not that easy to digest”, especially for readers unfamiliar with the ODE framework and physical modeling. Please see R1 for Reviewer Desm for the clarification of our goal and methodology contributions.
>
> The diffusion and reaction processes are commonly used physical models.  In our work, the diffusion process is to incorporate the correlation between the objects/entities from their graph structure in estimating their embedding trajectories. Accordingly, we can capture and leverage the objects’ underlying commonalities to bolster the learning. The reaction process is to model the individual evolution of the embeddings for each particular object, and hence it can capture the individual differences, i.e., personality. By combining the two processes, our model can capture both the commonalities and personalities among the objects to better learn their embedding trajectories. Our experimental results, as shown in Fig. 3, have confirmed that using the two processes jointly is advantageous over each single process. In our paper (see the last four lines of the 2nd paragraph in Page 4 and the text under Eq2), we have given the explanation from the physical perspective.
>
> We will provide more detailed explanations and discussions to improve the clarity and readability of our paper.
>
> C2: For the results in Table 1, it is entirely unclear that what does the RMSE error reflect? On pp.9, you only mention use remaining for test. But, I cannot find what are you predicting in the test set!
>
> R2: Thanks for the question. First, according to our problem setting (see the first paragraph of Section 2), we assume we observe a set of interactions, their results and timestamps.  An interaction is indexed by a tuple $\boldsymbol{\ell} = (l_1, \ldots, l_K)$ where each $l_k$ is the index of the $k$-th participant  entity. The interaction result can be payment (between customers, stores and products), price, count, etc. The dataset is denoted by $\mathcal{D} = \\{(\boldsymbol{\ell}_n, y_n, t_n)|n=1, \ldots, N\\}$, where $y_n$ and $t_n$ are the result and timestamp of $n$-th interaction $\boldsymbol{\ell}_n$, respectively.
>
> As stated in Page 9 (the last 4 lines in the first paragraph), we randomly sampled 80\% observed interaction results and timestamps (i.e., 80\% items in $\mathcal{D}$) for training, and used the trained model to predict the results of the remaining (20\%) interactions at their corresponding timestamps. For example, given a test interaction, say  $\boldsymbol{\ell}_8$, and the timestamp $t_8$, we use the trained model to predict the corresponding interaction result for which the ground-truth is $y_8$. Accordingly, we can calculate the RMSE. We repeated this experiment for five times and computed the average RMSE and standard deviation. We report the result for every method in Table 1. As mentioned in the paper, this is a standard setting used in tensor decomposition, which we followed to conduct our test. We will add more details to improve the clarity.

---

### Official Review · Reviewer_Desm · 2022-10-25

**Confidence:** 3
**Correctness:** 3
**Technical Novelty And Significance:** 3
**Empirical Novelty And Significance:** 3
**Recommendation:** 5

**Clarity, Quality, Novelty And Reproducibility:**

The authors provide quite enough detail about their work. It would be better if the abstract and the introduction can be refined.
Considering the work of [1], I’m not sure about the novelty of this paper.

**Strength And Weaknesses:**

Strengths

(1) This paper provides sufficient details in both theoretical and experimental sections. Although the authors didn’t release their code, I think it’s not hard to implement their work.

(2) The experimental results seem quite encouraging compared with the current baselines.

Weaknesses

(1) I suggest the authors reorganize this paper carefully. The author should emphasize their contributions concisely and accurately. In the current version, both the abstract and the introduction are filled with too many details and make me hard to capture what are the key contributions that the authors would like to present. It would be even better if the author could add a figure to demonstrate the architecture of the proposed framework.

(2) Could the authors discuss the correlations and differences with the work of [1]? It seems to have quite similar ideas with this paper, both the motivations and the methodologies. The authors should at least take it as a baseline.

[1] Li, Shibo, Robert Kirby, and Shandian Zhe. "Decomposing Temporal High-Order Interactions via Latent ODEs." In International Conference on Machine Learning, pp. 12797-12812. PMLR, 2022.

(3) The authors mention in the conclusion the proposed method is limited to a small number of entities. I suggest the authors provide some discussions about the algorithm complexity. Although the authors plan to develop graph cut algorithms to improve the scalability, I wonder how effective this algorithm can be.

(4) Typos: The last line of page 1, “these” should be “this”.

**Summary Of The Paper:**

This paper targets modeling the temporal high-order interactions via neural diffusion-reaction processes.
It proposes a Dynamical eMbedIngs of TempoRal hIgh-order interactions (DMITRI) framework.
A multi-partite graph is built to encode the correlation between the entities.
A graph diffusion process is constructed to co-evolve the embedding trajectories of the correlated entities and a neural network is used to capture the reaction process for each individual entity.

**Summary Of The Review:**

Considering [1], I’m not sure about the theoretical contributions and the results are SOTA.

---

> ### Author Response · Authors · 2022-11-13
> **Thanks for your valuable comments and suggestions.**
>
> C1:…The author should emphasize their contributions concisely and accurately. In the current version, both the abstract and the introduction are filled with too many details and make me hard to capture what are the key contributions that the authors would like to present. It would be even better if the author could add a figure to demonstrate the architecture of the proposed framework.
>
>
> R1: Thanks for your suggestion. We will polish our paper to better highlight the contributions. To be clear, here we briefly summarize our goal and methodology contributions.
>
> * Goal: We aim to estimate the **embedding trajectories** from temporal high-order interaction data among multi-type entities/objects (see Section 2, first paragraph).
>
> * Model: We use the ODE framework to construct a neural diffusion-reaction process model. The ODE state represents the continuous-time embedding of these objects. The diffusion process uses a multi-partite graph (constructed from data) to co-evolve the trajectories of neighboring nodes (objects), so as to take advantage of the graph correlation knowledge. The reaction process models the individual evolution of each object. The combination of the two processes (Eq4) can therefore capture both the commonalities and personalities in learning the embedding trajectories for different objects.
>
> * Algorithm: We use the ODE solver to develop a stochastic learning algorithm. We develop a stratified sampling approach to balance the cost of calling ODE solvers in each mini-batch so as to bolster the efficiency (Section 4).
>
> Actually, Fig. 1 of our paper has illustrated the key idea of our paper, where under a simple graph structure, the red arrows show that the diffusion process will jointly govern the embedding evolution of neighboring objects. The purple circle on each node indicates the individual reaction process that concentrates on learning the ``personality’’ of each object’s own trajectory. Together these correspond to Eq4 --- our neural diffusion-reaction embedding trajectory model. Given the embedding trajectories, we  use another neural network to generate the observed interaction results at different timestamps (Eq5). We have updated Fig. 1 at [here](https://github.com/anonymousiclr23/EmbeddingTrajectoryLearning/blob/main/vision-2-trim.pdf) to improve the clarify of our whole methodology. We will add this figure in our paper.

---

> > ### Author Response · Authors · 2022-11-13
> > **Response continue**
> >
> > C2: Could the authors discuss the work of Li et al. 2022 (reference [1])? It seems to have quite similar ideas with this paper, both the motivations and the methodologies. The authors should at least take it as a baseline.
> >
> > R2: Great question and suggestion! While the work of Li et al. 2022 also uses an ODE framework, the goal (motivation), the model design, and algorithm development are *all* very different.
> >
> > First, the goal of Li et al. 2022 is to estimate static (time-invariant) embeddings, which is the same as traditional tensor factorization.  The goal of our work, instead, is to learn **embedding trajectories**, i.e., a time function of the embeddings, to capture the temporal evolution of the objects’ representation.
> >
> > Second, Li et al. 2022 modeled **the result of each particular interaction** as an ODE, where the ODE dynamics (i.e., time derivative) is parameterized by a neural network and the (static) embedding representations of the involved objects. Such ODE can be viewed as a straightforward extension of the black-box neural ODE model. There is **not** any diffusion process. Overall, Li et al. 2022 created a set of ODEs, each for a particular interaction observed in data. By contrast, our work models the **embedding trajectories** as an ODE. There is only one ODE, whose state consists of the embedding trajectories of all the entities (in the data). Our ODE consists of two processes, one is a diffusion process that incorporates a multi-partite graph (constructed from the observed interactions) to co-evolve trajectories of the neighboring objects, and the other is an individual reaction process on each object. Both processes have physical interpretations (see the last four lines of the 2nd paragraph in Page 4 and the text under Eq2), which we believe are reasonable to apply to the embedding trajectory modeling.
> >
> > Third, for model estimation, the work of Li et al. 2022 uses forward sensitivity analysis to construct a companion ODE and compute the gradient by jointly solving the companion ODE. It also uses a time alignment method to solve all the ODEs to the same ending time. By contrast, our method uses the computational graph of the ODE solvers to compute the gradient. Since our model only has one single ODE, there is no time alignment issue, and we have to obtain the solutions at all the (distinct) time points in each mini-batch. We therefore developed a stratified sampling approach to balance the cost of the ODE solver in handling each mini-batch, so as to obtain an overall high learning efficiency.
> >
> > It is a great suggestion to also compare with (Li et al. 2022) that is named as THIS-ODE in the original paper. Here we add the comparison results.
> >
> > * *CA Weather* dataset
> > |               |                          |                                  |                       |                           |
> > | :---------: | :----------------: | :----------------------: | :--------------: |  :----------------: |
> > | *Method*       |       $R=2$    |             $R=3$      |          $R=5$ |  $R=7$ |
> > | THIS-ODE      | $0.7511	 \pm 0.0052$|$	0.7539 \pm	0.0041$|$	0.7614 \pm	0.0024$|$	0.7620 \pm	0.0032$|
> > | DMITRI   | **0.6327** $\pm$	**0.0119**|	**0.6109**	$\pm$ **0.0056**	|**0.6172** $\pm$ **0.0075**|**0.6354**$\pm$ **0.0085**|
> > |               |                          |                                  |                       |                           |
> >
> >
> > * *CA Traffic* dataset
> > |               |                          |                                  |                       |                           |
> > | :---------: | :----------------: | :----------------------: | :--------------: |  :----------------: |
> > | *Method*       |       $R=2$    |             $R=3$      |          $R=5$ |  $R=7$ |
> > | THIS-ODE      | $0.6603 \pm	0.0230$|$	0.6536	\pm 0.0212$|$	0.6838 \pm	0.0193$|$	0.6378\pm	0.0142$|
> > | DMITRI   | **0.3601** $\pm$	**0.0334**|	**0.2972** $\pm$	**0.0099**|	**0.3174** $\pm$	**0.0118**| **0.3269** $\pm$ **0.0162**|
> > |               |                          |                                  |                       |                           |
> >
> > * *Server Room* dataset
> > |               |                          |                                  |                       |                           |
> > | :---------: | :----------------: | :----------------------: | :--------------: |  :----------------: |
> > | *Method*       |       $R=2$    |             $R=3$      |          $R=5$ |  $R=7$ |
> > | THIS-ODE      | $0.1412	 \pm0.0024$|$	0.1312\pm	0.0013$ | $	0.1304 \pm	0.0016$|$	0.1350 \pm	0.0019$|
> > | DMITRI   | **0.0536** $\pm$	**0.0031**| **0.0403** $\pm$ **0.0014**| **0.0393** $\pm$ **0.0018**| **0.0403** $\pm$ **0.0027**|
> > |               |                          |                                  |                       |                           |
> >
> > As we can see, our method, DMITRI, consistently outperforms THIS-ODE. We will supplement all these discussions and results into our paper.

---

> > > ### Author Response · Authors · 2022-11-13
> > > **Response continue**
> > >
> > > C3: The authors mention in the conclusion the proposed method is limited to a small number of entities. I suggest the authors provide some discussions about the algorithm complexity. Although the authors plan to develop graph cut algorithms to improve the scalability, I wonder how effective this algorithm can be.
> > >
> > > R3: Great suggestion and question. The time complexity of our method is $\mathcal{O}((M+d)RT)$ where $M$ is the number of non-zero entries (i.e., edges) in the stacked graph-Laplacian matrix $\mathcal{W} -\mathcal{A}$ in Eq4, and $d = d_1 + \ldots + d_K$ is the total number of entities in data, $R$ is the dimension of the embeddings, and $T$ is the number of numerical integration steps. Since we do not need to store the entire history of the numerical integration, the space complexity is $\mathcal{O}(M + dRS)$, which is to store $\mathcal{W}-\mathcal{A}$ and the embedding trajectory values at the current step and a few previous steps necessary for numerical integration (in total $S$). Note that when the graph is dense, $M = \mathcal{O}(d^2)$. We will supplement the complexity analysis in our paper.
> > >
> > > Since our ODE model combines the embedding trajectories of all the entities into one single ODE to fulfill the diffusion process on the entire graph. When the number of nodes ($d$) is large, the graph will become very large and $M$ can be large as well.  Consequently, the learning with the single ODE can be prohibitively costly. However, if we can use graph cut algorithms to partition the giant graph into a set of much smaller subgraphs where the nodes of each subgraph do not overlap, then different subgraphs can be viewed as disconnected to each other. Hence, we only need to construct an ODE for each subgraph and the diffusion process is only within that subgraph. On each subgraph, the number of nodes ($d$) and edges ($M$) are much smaller. Accordingly, each ODE is at a much smaller scale and much more efficient for computation. It also brings a possibility to parallelize the model estimation. We will investigate these ideas in the future work.

---

### Official Review · Reviewer_cuTX · 2022-10-25

**Confidence:** 3
**Correctness:** 3
**Technical Novelty And Significance:** 3
**Empirical Novelty And Significance:** 2
**Recommendation:** 8

**Clarity, Quality, Novelty And Reproducibility:**

- Clarity: High. I did have any problems understanding the paper.
- Quality: High. The proposed DMITRI framework appears technically sound along with the experiments.
- Novelty: High. I have not previously seen neural ODEs used to model multi-entity interactions. Additionally, continuous-time models in this area are rare yet potentially more powerful than discrete-time models.
- Reproducibility: Low. No code is provided, and the experiments use sampled subsets of larger data sets, e.g. 10K interactions from the Server Room data. Without knowing how the 10K interactions were selected, it would be difficult to reproduce results.

**Strength And Weaknesses:**

Strengths:
- Proposed DMITRI framework generates dynamic embeddings for entities in continuous, rather than discrete time, unlike most existing dynamic tensor factorization approaches.
- Use of neural ODEs to generate embeddings for multi-entity interactions appears to be novel.
- Comparisons to a wide range of competing methods yields impressive predictive accuracy on 3 real datasets.

Weaknesses:
- Minimal interpretations of the learned embedding trajectories. Beyond predicted observations, an additional output of DMITRI is the continuous-time embedding trajectories. The authors don't provide many interpretations of the trajectories, with only an example in Section 6.3.
- No empirical results on computation time. The authors state that their approach is limited to a small number of entities due to the multi-partite graph approach for the diffusion process. I would have liked to see some experiments (perhaps in the supplementary material) comparing wall clock time to competing methods.

Question:
1. On page 4, you state that there are $K(K+1)/2$ adjacency matrices $\mathbf{W}^{k,k'}$. Why $K(K+1)/2$ rather than $K(K-1)$? It appears that we can't have edges within the same type $k$, so no diagonal, and that there is no symmetry between $\mathbf{W}^{k,k'}$ and $\mathbf{W}^{k',k}$.

Typo:
- Page 6, second last paragraph: Unifrom\[0,5\] -> Uniform\[0,5\]

**Summary Of The Paper:**

The authors propose the DMITRI framework to generate dynamic embeddings for multi-entity interactions with timestamps. A multi-entity interaction involves potentially many types of entities, e.g. traffic type, severity, latitude, and longitude, and consists of a timestamped observation for those entities, such as the number of traffic accidents under those conditions at different times. The DMITRI framework uses an ordinary differential equation (ODE) model to learn embedding trajectories for each entity of each type. The authors consider both a diffusion process across multi-partite graphs, representing interactions between different types, and a reaction process for the individual entities. They demonstrate improvements in predictive accuracy on 3 real datasets compared to a large number of existing methods.

**Summary Of The Review:**

I find this to be a novel and interesting contribution with strong empirical results for prediction accuracy, but also a few weaknesses in the experimental evaluation that prevent me from strongly recommending acceptance.

---

> ### Author Response · Authors · 2022-11-13
> **Thanks for your insightful comments and suggestions**
>
> C1: … Minimal interpretations of the learned embedding trajectories.
>
> R1: Thanks for your great comments and we do agree. Please **see R2 for Reviewer U7s8**, where we provide a more detailed analysis about the learned embedding trajectories for location 1 of *Server Room* dataset.  We will add the analysis of *all* the trajectories shown in Fig. 4 in our paper.
>
> In addition, we have also added an investigation about the structures of the embeddings along with time, as shown [here](https://github.com/anonymousiclr23/EmbeddingTrajectoryLearning/blob/main/clustering.pdf). The results exhibit the evolution patterns of all the locations on *Server Room* dataset, which give a systematic view of the temporal structures within the entities. Please see **R2 for Reviewer U7s8** for more detailed discussion. Again, we will add these results and analysis into our paper.
>
> C2: No empirical results on computation time … I would have liked to see some experiments comparing wall clock time to competing methods.
>
> R2: Thanks for the great suggestion!  Here, we supplement the per-epoch running time (in seconds) of each method. We tested all the methods  in a workstation with one NVIDIA GeForce RTX 3090 Graphics Card, 10th Generation Intel Core i9-10850K Processor, 32 GB RAM, and 1 TB SSD. We can see that the running speed of our method is comparable to NONFAT, the most recent work for embedding trajectory learning. We will add these results into our paper (supplementary material).
>
> |               |                          |                                  |                       |
> | :---------: | :----------------: | :----------------------: | :--------------: |
> | *Method*       |      *CA Weather*    |            *CA Traffic*     |   *Server Room* |
> |CP-DTLD |0.037| 0.086| 0.023|
> |GP-DTLD	|0.246| 0.247 |0.248|
> |NN-DTLD	|2.400| 4.730 |1.080|
> |CP-DTND	|0.038 | 0.087 |0.025|
> |GP-DTND |0.119 |0.242| 0.080|
> |NN-DTND	|2.360| 4.701| 1.060|
> |CP-CT	|0.025 | 0.052 |0.018|
> |GP-CT	|0.068| 0.216 |0.105|
> |NN-CT	|2.310| 3.885 |1.030|
> |NONFAT|	0.952| 1.925| 0.571|
> |DMITRI	|1.390 |1.895| 0.309|
> |               |                          |                                  |                       |
>
> C3: Regarding the number of adjacent matrices. Why rather than $K(K-1)$?
>
> R3: Thanks for the great catch! It is a typo, and the number of adjacent matrices is indeed $K(K-1)$. We will correct the typo in our paper.
>
> C4: Reproducibility concern.
>
> R4: Great comments! We have shared our code and data (anonymously) in Github at [here](https://github.com/anonymousiclr23/EmbeddingTrajectoryLearning).

---

> > ### Comment · Reviewer_cuTX · 2022-12-02
> > **Thank you for the responses**
> >
> > Thank you for the responses! My concerns have mostly been addressed, and I continue to support the paper.

---

### Official Review · Reviewer_U7s8 · 2022-10-29

**Confidence:** 3
**Correctness:** 3
**Technical Novelty And Significance:** 2
**Empirical Novelty And Significance:** 2
**Recommendation:** 6

**Clarity, Quality, Novelty And Reproducibility:**

ODE-based approach has been studied, but the authors extend it to multi-way interactions. Even though it is an interesting extension, the proposed method does not have a significant change beyond the related work.

Manuscript needs to be presented better. For example, ODE explanation has lots of repeated explanation, which distracts the main model description.

**Strength And Weaknesses:**

* Strengths
- ODE is a reasonable way of representing the dynamics of embeddings.
- Authors use both synthetic and real-world datasets and show the proposed method can track the temporal patterns well.
- Empirical evaluations show that the proposed method outperforms the other baseline models.

* Weaknesses
- The manuscript can be further improved to present better. Crucial information -- such as the experimental task setup -- or explanation is often not very clear.
- The authors verify the end-to-end predictive power, but it would be great if the authors could analyze the insights from the dynamics of embeddings.
- For the dynamic embedding setting, it would be great to have some forecasting tasks for evaluations.
- It would be great if the authors evaluate on the other kinds of data, where the actual events are sparse like social interaction datasets. The provided datasets have the characteristics that graphs are generated from different forms of dataset, not naturally given such as social interactions or citations.

**Summary Of The Paper:**

Authors address the problem of temporal high-order interactions where more than two entities -- such as customers, commodities, and merchants -- correspond to one event. To tack the temporal drift of underlying node, authors propose the dynamic embeddings that follows a diffusion-based ODE model. Authors show that the proposed DMITRI model can capture the temporal dynamic well through both synthetic and real-world datasets.

**Summary Of The Review:**

While the proposed method is incremental to existing work, authors translate the states over multi-dimensional space into a graph among multi-entities, apply the extended ODE, and show the good performance. The proposed method is worth being discussed in the community.

---

> ### Author Response · Authors · 2022-11-13
> **Thanks for your valuable comments and suggestions.**
>
> Thanks for your valuable comments and suggestions. Here are our responses. C: comments; R: response.
>
> C1: … The manuscript can be further improved to present better. e.g., crucial information -- such as the experimental task setup -- is often not very clear; …ODE explanation has lots of repeated explanation, which distracts the main model description.
>
> R1: Great suggestions! We will shrink the presentation of the ODE model to make it more concise. We will also supplement more details about the experimental setting to improve the clarity.
>
> Here, we briefly highlight the task setup. In the ablation study (Section 6.1), we generated a synthetic dataset according to some ground-truth embedding trajectories (detailed in the first paragraph of Section 6.1). Then we ran our method and the competing method NONFAT on that synthetic dataset to recover the embedding trajectories. As shown in Fig. 2, our method recovered the trajectories and their structures quite accurately while NONFAT did not, although they achieved very close prediction accuracy. In Section 6.2, we examined our method and 10 competing methods in predicting the interaction results at different timestamps. We randomly collected 80% of observed interactions and their timestamps to train each model and used the trained model to predict the remaining 20% interaction results (at the corresponding timestamps). We repeated this procedure for five times and reported the average root-mean-square error (RMSE). As stated in our paper, this is a standard setting for tensor decomposition of high-order interactions used by prior works, which we followed to conduct our evaluation (see Page 9, 1st paragraph, last four sentences). We will highlight this point in our paper.
>
>
> C2: The authors verify the end-to-end predictive power, but it would be great if the authors could analyze the insights from the dynamics of embeddings.
>
> R2: Thanks for the great suggestion! In Fig. 4 (Section 6.3), we have looked into the learned trajectories for the entities in *Server Room* dataset. We can see that the trajectories even for the same entity (like one location) can vary quite differently. For example, Fig. 4a, b, and c are the three trajectories for location 1. Fig. 4a exhibits a negative quadratic pattern (i.e., increasing first, and then decreasing). Fig. 4b first decreases and then increases.  Fig. 4c keeps decreasing with time. These trajectories can reflect different time-varying properties of that entity (e.g., the workload, memory usage at that location). Note that since the data does not contain any additional information about the entities (we only have the interaction results and timestamps among the objects/entities), it is hard to tell which particular properties these trajectories represent and/or summarize. But the results do show some interesting patterns.
>
> In addition, we add an investigation about how the structures of the embeddings vary along with time, which are shown [here](https://github.com/anonymousiclr23/EmbeddingTrajectoryLearning/blob/main/clustering.pdf) .  We looked into the embeddings of the 34 locations on *Server Room* dataset at five time points (t = 1, 20, 50, 80, 100). At each time, we ran the k-means algorithm over the embeddings to extract the clustering structures. We used the elbow method [1] to select the number of clusters.  We can see that at earlier time (t<=50), the clusters are more compact, while at the later stages, the clusters become more scattered. This reflects how the structure of those entities (i.e., locations) evolves along with time. It is interesting to see that some locations are in the same cluster all the time, like location {5,7} and location {16, 32}.  It implies that their underlying properties might have quite similar (or correlated) evolution patterns.  Some locations are grouped in the cluster at the beginning, e.g., location {32, 34} (at t = 1), but later moves to different clusters (t>1). It implies their evolution patterns can vary significantly, leading to the change of the cluster memberships.
>
> We will supplement all these analysis and additional results into our paper.
>
> [1] Ketchen, David J., and Christopher L. Shook. "The application of cluster analysis in strategic management research: an analysis and critique." Strategic management journal 17.6 (1996): 441-458.

---

> > ### Author Response · Authors · 2022-11-13
> > **Response continue**
> >
> > C3: For the dynamic embedding setting, it would be great to have some forecasting tasks for evaluations.
> >
> > R3: Great suggestion! Since we followed the test setting prevalent in the tensor decomposition, the training and test interactions as well as their timestamps are randomly sampled. Hence, the timestamps are mixed, and the test timestamps can be smaller than (some) training timestamps.
> >
> > Here, we supplement an experiment about the forecasting task. Specifically, on the *Server Room* dataset, we first sorted the observed interactions according to their timestamps. We used the first 10K interactions as the training pool and used the last 5K interactions as the test pool. Then we randomly sampled 8K interactions from the training pool to train each model, and randomly sampled 2K interactions from the test pool as the test set. We evaluated the predictive error of each model on the test set. Therefore, the timestamp of each test interaction is guaranteed to be larger than all the timestamps of the training interactions. We repeated the experiment for five times, and report the average RMSE and standard deviation in the following Table. As we can see, our method DMITRI outperforms all the competing methods, which is consistent with Table 1 in our paper. We will add these results into our paper.
> >
> > |                |                           |                                   |                        |                            |
> > | :---------: | :----------------: | :----------------------: | :--------------:  |   :----------------: |
> > | *Method*        |        $R=2$     |              $R=3$       |           $R=5$ |   $R=7$  |
> > | CP-DTLD       | $ 0.1969 \pm 0.0030$ | $0.1943 \pm	0.0022$ |	$0.1949 \pm 0.0023$ | $ 0.1945 \pm	0.0021$|
> > | GP-DTLD | $0.0627 \pm 0.0012$ | $0.0595\pm	0.0020$ | $0.0635\pm 0.0010$ | $0.0623\pm 0.0013$|
> > | NN-DTLD | $0.2092 \pm 0.0059$ | $0.1996 \pm	0.0026$ | $0.1945\pm 0.0023$ | $0.1929 \pm 0.0031$|
> > | CP-DTND | $0.2694 \pm 0.1594$ | $0.1494	\pm 0.0014$ | $0.1793 \pm 0.0629$| $0.2778 \pm 0.0816$|
> > |GP-DTND |  $0.0633 \pm 0.0018$ | $0.0598\pm	0.0019$ | $0.0635\pm 0.0012$ | $ 0.0634 \pm 0.0013$|
> > |NN-DTND |  $0.2041 \pm 0.0019$ | $0.2001 \pm	0.0022$ | $0.1983 \pm 0.0024$ | $ 0.1967 \pm 0.0020$|
> > |CP-CT      |  $0.9594 \pm 0.0127$ | $0.9283 \pm	0.0106$ | $0.9782 \pm 0.0083$ | $0.9645 \pm 0.0111$|
> > |GP-CT      | $0.0725 \pm 0.0003$ | $0.0741 \pm	0.0011$ | $0.0999 \pm 0.0076$ | $0.1398 \pm 0.0013$|
> > |NN-CT      | $ 0.7048 \pm 0.0105$ | $0.7241 \pm	0.0043$ | $0.7193\pm 0.0066$ | $0.7224 \pm 0.0045$|
> > | DMITRI    | **0.0478** $\pm$ **0.0012**  | **0.0460** $\pm$ **0.0017** | **0.0451** $\pm$ **0.0019** | **0.0435** $\pm$ **0.0037**|
> > |                |                           |                                   |                        |                            |
> >
> > C4: It would be great if the authors evaluate on the other kinds of data, where the actual events are sparse like social interaction datasets. The provided datasets have the characteristics that graphs are generated from different forms of dataset, not naturally given such as social interactions or citations.
> >
> >
> > R4: Great suggestion! while our method can straightforwardly incorporate additional data source for graph construction, that might be unfair when comparing with the baseline methods which cannot do so. Hence, in our experiments, we only used the same dataset ("from different forms") to generate the graphs, so as to make sure the improvement of our method does not come from additional information source. However, we **do agree** your suggestion. We will surely find out such "social interaction or citation datasets" for additional evaluations and to further enhance our paper.

---

### Author Response · Authors · 2022-11-19
**Summarizing our response**

We would like to thank the reviewers for the valuable suggestions and insightful comments. We believe our responses to each reviewer have addressed most of their concerns. We will integrate the explanations, discussions, and supplemented results in the below into our paper.

Hence, we would appreciate if the reviewers could please reconsider their assessment of our work based on our responses, or alternatively let us know if they still have concerns that we could try to address further.

In summary, here are in general items what we addressed (and where we addressed):

**Clarification regarding the experimental setting**. We described to reviewer eUMJ (in R2) that our experiments followed the standard tensor decomposition setting (as referenced by our paper), which used 80% observed data for learning (the embedding trajectories), and then predicted the interaction results of the remaining 20% interactions at corresponding timestamps. We gave a more detailed explanation based on the mathematical notations of the data used in our paper. We also summarized this procedure and the synthetic task setting to Reviewer U7s8 (in R1).

**Interpretation about the learned embedding trajectories**. We provided more detailed analysis of the learned trajectory embeddings shown in our paper. In addition, we added a study about the dynamic clustering structures of the entities according to their embedding trajectories (see [here](https://github.com/anonymousiclr23/EmbeddingTrajectoryLearning/blob/main/clustering.pdf) ). See R2 for Reviewer U7s8 and R1 for Reviewer cuTX.

**Clarification about the model design and difference with the recent work (Li et al., 2022)**.  We thank Reviewer Desm for pointing out the most recent work (Li et al., 2022) that also uses an ODE framework to handle temporal interaction data. We explained this work is totally different from our method in the goal (motivation), modeling, and algorithm. We have also supplemented the comparison results with (Li et al., 2022), which show that our method consistently achieves better prediction accuracy. See R2 for Reviewer Desm.

In addition, we summarized and highlighted our methodology contributions. We updated the model figure [here](https://github.com/anonymousiclr23/EmbeddingTrajectoryLearning/blob/main/vision-2-trim.pdf) to better highlight the model architecture. See R1 for Reviewer Desm and R1 for Reviewer eUMJ.

**Forecasting Task**. We supplemented the results of a forecasting task, according to the suggestion of Reviewer U7s8 (see R3). Our method shows consistent advantages over the competing approaches.


**Reproducibility**. We shared our code and data anonymously in Github [here](https://github.com/anonymousiclr23/EmbeddingTrajectoryLearning).

**Running time, complexity analysis, and future work**. We provided the running time comparison (see R2 for Reviewer cuTX). We also described to Reviewer Desm the algorithm complexity, based on which we explained why our idea of using graph cutting algorithms to split the giant graph into many small sub-graphs could possible improve the scalability of our method (see R3 for Reviewer Desm).

---

### Decision · Program_Chairs · 2023-01-20

**Decision:**

Reject

**Justification For Why Not Higher Score:**

See the weaknesses discussed above.

**Justification For Why Not Lower Score:**

N/A

**Metareview: Summary, Strengths And Weaknesses:**

In this paper, the authors considered a time-varying tensor decomposition scenario and proposed an ODE-based method called DMITRI to capture the dynamics of latent embeddings. The proposed method considers the high-order interactions among different entities over time and leverages a diffusion-reaction process to model the evolution of the embeddings. Experiments demonstrate the usefulness of the proposed method.

Strengths:
(1) The proposed method is reasonable, and its derivation is detailed and clear.

Weaknesses:
(1) The novelty of the proposed method is limited. Many temporal tensor factorization methods have been proposed. Additionally, in the continuous-time cases, point process (PP)-based factorization methods should also be discussed and considered as baselines. Actually, for recent years, there are some ODE/SDE-based sequential models embedding evolutionary entities. The differences and the connections between them and the proposed method are not clarified.
(2) The experiments are insufficient. As aforementioned, some PP-based methods and SDE-based sequential models should be considered.

**Summary Of Ac-Reviewer Meeting:**

After the rebuttal phase, the scores of the submission are 6, 8, 5, 5. Except for the reviewer giving 8, three of the four reviewers have concerns about the novelty of the proposed method. I think the idea of the proposed method is interesting, but I agree with the reviewers that the differences between the proposed method and other related work (point processes and SDE-based method) and the advantages of the proposed method should be further discussed. I did ask for a virtual meeting, but the reviewers were not responsive.
According to the analysis above, I tend to reject this work.